# Time Course of Reactive Brain Activities during a Stroop Color-Word Task: Evidence of Specific Facilitation and Interference Effects

**DOI:** 10.3390/brainsci13070982

**Published:** 2023-06-22

**Authors:** Francesco Di Russo, Valentina Bianco

**Affiliations:** 1Department of Movement, Human and Health Sciences, University of Rome “Foro Italico”, 00135 Rome, Italy; francesco.dirusso@uniroma4.it; 2Santa Lucia Foundation IRCCS, 00179 Rome, Italy; 3Department of Brain and Behavioural Sciences, University of Pavia, 27100 Pavia, Italy

**Keywords:** Stroop, ERP, facilitation, interference

## Abstract

The Stroop test represents a widely used task in basic and clinical research for approaching the cognitive system functioning in humans. However, a clear overview of the neurophysiological signatures associated with the different sub-domains of this task remains controversial. In the present study, we leveraged the EEG technique to explore the modulation of specific post-stimulus ERPs components during the Stroop test. Critically, to better disentangle the contribution of facilitation (i.e., faster color identification times for color-congruent Stroop words) and interference (i.e., longer color identification times for color-incongruent Stroop words) processes prompted by the Stroop test, we delivered congruent and incongruent trials in two separate experimental blocks, each including the respective neutral condition. Thanks to this methodological manipulation, we were able to clearly dissociate the two sub-processes. Electrophysiological results suggest specific markers of brain activity for the facilitation and the interference effects. Indeed, distinctive Stroop-related ERPs (i.e., the P3, the N450, and the LPC) were differently modulated in the two sub-processes. Collectively, we provide evidence of selected brain activities involved in the reactive stage of processing associated with the Stroop effect.

## 1. Introduction

Our executive system must continuously process efficiently a large flow of different information from the external environment. To avoid overloads, the brain voluntarily processes only a portion of these features while anything else is processed automatically. This dual architecture of brain cognition is managed by the cognitive system, which acts in a goal-directed and flexible fashion, thus allowing us to resist automatic tendencies [1]. This depends on our ability to handle distracting stimuli, which can impair the goal. In other words, we must be able to inhibit the cognitive interference occurring when the processing of a stimulus feature affects the simultaneous processing of other features of the same percept.

One common way to study the effect of cognitive interference on performance is the Stroop test [2], in which participants must quickly identify one single object feature, ignoring the others. The Stroop color and word test (SCWT) is the largely used version for both experimental and clinical purposes. In the SCWT, participants are usually required to detect, as fast as possible, the ink color in a sequence of color words. If the word and the ink color are congruent, the performance is high, but, if the two features are incongruent, performance is low because word reading is mostly automatic and it is difficult to inhibit the irrelevant word meaning, thus giving rise to the “Stroop effect” [3]. This paradigm is largely used to challenge flexible behavior and to provide a measure of cognitive inhibition to overlearn dominant responses [4,5]. Considerable interest in the Stroop task lies in its utility as a prominent tool to probe cognitive functions both in basic research and clinical settings [6,7,8].

A critical debate in Stroop’s literature regards the locus of cognitive interference in the SCWT. Some views claim that this occurs at early stimulus processing stages. For instance, the “parallel distributed processing” model [9] posits that word and color processing occur in parallel, along pathways of different strengths; then, in light of the augmented reading practice, attention would reinforce the strength of the colored word. Therefore, the information processing capabilities of the pathway would influence the probability of the response production response [10,11]. In contrast, the late selection response competition theories propose that the locus of interference in the SCWT occurs later at the output stage and is explained as response competition [12,13].

Event-related potentials (ERPs), leveraging the high temporal resolution, represent a suitable technique to address the debate between early and late processing stages of cognitive interference. However, most Stroop studies using ERP not only are outdated but also present conflicting results. In brief, some studies supported the idea of cognitive interference acting at early stages of perceptual and attentional processing between 250 and 600 ms post-stimulus [14,15,16,17]. Other studies supported late selection explanations just before response emission [9,18,19,20,21]. Collectively, these studies describe the modulation of Stroop effects on three main ERP components: the P3, an ERP emerging between 300 and 700 ms post-stimulus, which is typically highest on centroparietal areas and whose amplitude mainly represents the amount of attention used during stimuli processing [22]; the N450, a frontocentral-distributed component with onset between 400 and 600 ms post-stimulus and reflecting conflict detection [23]; and the late positive complex (LPC), maximal at 600 ms post-stimulus, indexing contextual integration and post-decision closure [24].

Further, as highlighted in a recent review by Heidlmayr and co-authors [25], ERPs represent the ideal methodological tool for tracing with high precision the time course of different executive sub-processes involved in conflict monitoring, interference suppression, and conflict resolution in the Stroop task. Relatedly, the authors acknowledged the existence of a centro-posterior N400 (reflecting inhibitory processes and interference suppression) and, subsequently, a late sustained potential (LSP; discussed as reflecting either an engagement of executive processes [26], conflict resolution processes [27,28,29,30], semantic reactivation of the meaning of words following conflict resolution [20,31], or response selection [32].

Another caveat in Stroop’s literature is that several studies compared congruent and incongruent conditions [33] but did not include a neutral condition (e.g., a non-color word such as BOOK written in red) in the task, preventing the dissociation of facilitation and pure interference effects. The use of neutral conditions is important to have a control condition to properly dissociate the effects related to incongruence from those related to congruence. Further, when neutral stimuli in the Stroop task consisted of pseudowords, which do not induce lexical activations, the discussion with word-color conditions (both congruent and incongruent) cannot exclude the confounding effect t of lexicality. For instance, Burt [34] showed quicker color naming of neutral words than nonwords, indicating that the effect is indeed linked to lexical activation.

This ERP investigation attempts to acknowledge new evidence to address the existing debates among contrasting models of the Stroop test. Specifically, we sought to test the early and late selection hypothesis using neutral words, which were separately intermixed with congruent or incongruent words to obtain a reliable control condition for each category. In this way, we could disentangle within the Stroop effect pure interference generated by incongruency and facilitation induced by the congruent color words. In light of the novelty of our experimental design in relation to existing ERP literature addressing the Stroop effect, we opted for an exploratory approach concerning the occurrence of specific ERP components. However, we hypothesized that the components mainly related to conflict monitoring and/or conflict resolution were higher in amplitude in the more control-demanding condition compared to the less control-demanding one.

## 2. Methods

### 2.1. Participants

A total of 18 right-handed adults (8F, 26 ± 6.8 years) participated in this experiment. The sample size was determined through G*POWER software [35]. On the basis of the findings of a previous ERP study on the Stroop test [36], a medium effect size of f(U) = 0.77 was estimated. The significance level was set at α = 0.05, and the desired power (1−β) was set at 0.95. Participants were all university students recruited at the University of Foro Italico with a similar socioeconomic status. Inclusion criteria included age between 18 and 35 years, having normal or corrected-to-normal vision, the absence of any neurological or psychological disorders, being free from psychotropic medications, and being right-handed (Edinburgh Handedness Inventory, EHI, [37]). Written informed consent was obtained before starting the experiment. Participants were blind to the experimental aims and were debriefed only at the end of the experimental procedure.

### 2.2. Apparatus and Task Procedure

Presentation Software (Neurobehavioral Systems, Inc., Berkeley, CA, USA) was used to present experimental stimuli. These were Italian words presented in four possible ink colors (red, blue, green, and yellow), 0.5 cm from a white fixation cross in the center of a computer screen on a grey background. The words were taken from the Italian language, blue (blu), yellow (giallo), red (rosso), and green (verde), and could be congruent (i.e., green printed in green) or incongruent (i.e., green printed in red). Neutral words were time (tempo), epoch (epoca), hit (colpo), and rigid (rigido). These words were selected to match the length of the color words. The words were presented individually, in lower case (Arial font, size 36), and covered 1.00° of the visual angle horizontally and 0.30° vertically. The fixation cross had a diameter of 0.15° of the visual angle.

Participants were seated, and the computer was placed 114 cm from their eyes. They had to provide their response by means of a push button board with four buttons, which corresponded to the four-color alternatives. They were allowed to provide their choice using the index or the middle finger of the left and right hands. Instructions included maintaining the gaze on the fixation and responding ASAP by operating the button corresponding to the ink color of the delivered words. Words were present for 750 ms and the inter-stimulus interval (ISI) ranged between 1.5 and 2.5 s. Each run lasted 72 trials. In the “Congruent-Neutral” run (i.e., congruent trials intermixed with neutral trials), congruent and neutral trials were presented in a random fashion with a 0.5 probability. In the “Incongruent-Neutral” run (i.e., incongruent trials intermixed with neutral trials), incongruent and neutral trials were presented in a random fashion with a 0.5 probability. Each experimental session consisted of 6 of each ‘Congruent—Neutral’ and ‘Incongruent—Neutral’ runs, and the presentation order was randomized between runs and between participants. The neutral words being presented with both the congruent and incongruent color words in different bocks were defined as ‘Neutral—C’ and ‘Neutral—I’ conditions, respectively.

### 2.3. Behavioral Performance

The speed was measured using the individual mean response time (RT) for correct trials. The performance accuracy was measured using the commission errors percentage (%CE). RT and %CE were submitted to separate one-way ANOVAs with conditions (congruent, incongruent, Neutral-C, and Neutral-I) as independent variables. The percentage of omission errors (OE) was very low (<1%) and was not submitted for further analysis.

### 2.4. EEG

Subjects were individually tested in a dimly lit room using an 80-channel EEG system (Brainamp™ amplifier) with 64-channel electrodes (Acticap™) and software (Recorder 1.2) all by Brain Products GmbH (Munich, Germany). The scalp electrodes were mounted according to the 10–10 International System and initially referenced to the left mastoid. Horizontal and vertical electrooculograms (EOG) were measured by bipolar recordings. The EEG was digitized at 250 Hz, amplified (bandpass of 0.01–60 Hz including a 50 Hz notch filter), and stored for offline averaging. The removal of eye movement artifacts was performed using independent component analysis (ICA) ocular correction [38]. Data were high-pass- (30 Hz) and low-pass (0.1 Hz)-filtered, and semi-automatic artifact rejection was carried out before signal averaging to exclude epochs with signals exceeding the amplitude threshold of ±80 μV. Segmentation included epochs beginning 200 ms pre-stimulus and ending 1000 ms post-stimulus.

To control for multiple comparisons, we used the collapsed localizers method [39] to choose the regions of interest (ROIs) and the time windows for the following statistical analysis. Since we aimed to identify varying topographies independently by conditions (that were collapsed together), we inspected ERPs by looking at the global field power (GFP) at scalp mapping and considering previous similar ERP studies on the Stroop test [17,20]. We focused on differential activity to isolate the facilitation effects of the congruent trials over the neutral trials and the pure interference effects of the incongruent trials over the neutral trials associated with the Stroop effect. Three differential ERPs were obtained: (1) facilitation, congruent minus neutral in the congruent block; (2) pure interference, incongruent minus neutral in the incongruent block; and (3) spurious interference, incongruent minus congruent trials as performed in many previous studies. GFP analysis indicated three peaks, and, considering an interval within 80% of each peak, the following time windows were identified: 316–400 ms, 450–588 ms, and 600–856 ms. These intervals were, respectively, associated with the P3, N450, and LPC Stroop effects well described in the literature [20,24]. In these intervals, the electrodes with an amplitude within 80% of the peak electrode were polled in ROIs. The P3 and the LPC, having a similar scalp distribution, were represented by a medial parietal ROI (P1-Pz-P2-POz), and the N450 was represented by a medial frontocentral pool (FC1-FCz-FC2-Cz). All raw and differential component amplitudes were calculated with respect to the 200 ms pre-stimulus baseline. For each time window and relative ROI, a 3-level one-way ANOVA was performed on differential amplitudes with the Stroop effect (facilitation, pure interference, and spurious interference).

Post hoc comparisons were made using a Bonferroni-based test (dividing the *p*-value for the number of the used comparisons). The partial eta squared (_p_η^2^) was used to measure the effect size of the significant effects. The overall alpha value was fixed at 0.05.

## 3. Results

### 3.1. Behavioral Results

Table 1 includes the behavioral performance. The ANOVA showed a significant effect of the RT (F_3,51_ = 33.5, *p* < 0.001, _p_η^2^ = 0.663). The RT for incongruent trials was slower than RTs for all other conditions (*p* < 0.001). The RTs for Neutral-I trials were slower than RTs for congruent trials (*p* = 0.032) and for Neutral-C trials (*p* = 0.041). A significant effect of accuracy emerged (F_3,51_ = 12.1, *p* < 0.001, _p_η^2^ = 0.416). The CE was higher for incongruent trials than for all other conditions (*p* < 0.001). The CE for congruent trials was lower than for Neutral-C trials (*p* = 0.007) and Neutral-I trials (*p* = 0.001), which did not differ from each other.

### 3.2. Electrophysiological Results

Figure 1 shows the post-stimulus ERP obtained in each of the four conditions. Figure 2 shows the differential waves in the two considered ROIs showing the P3, N450, and LPC effects. Figure 3 shows the relative voltage and CSD distribution in the three studied intervals.

The ANOVA on the P3 effect was significant (F_2,34_ = 12.2, *p* < 0.001, _p_η^2^ = 0.418). Post hoc analyses showed that the amplitude of the P3 effect in the spurious interference condition (−0.19 µV) was smaller than the P3 effect in the pure interference (1.34 µV, *p* < 0.001) and facilitation (1.51 µV, *p* < 0.001) conditions, which did not differ from each other.

The ANOVA on the N450 effect was significant (F_2,34_ = 5.3, *p* = 0.010, _p_η^2^ = 0.262). Post hoc analyses showed that the amplitude of the N450 effect in the pure interference condition (−1.70 µV) was larger than in the facilitation condition (−0.34 µV, *p* = 0.007) but did not differ from the amplitude of the spurious interference condition (−0.88 µV).

The ANOVA on the LPC effect was significant (F_2,34_ = 4.0, *p* = 0.025, _p_η^2^ = 0.193). Post hoc analyses showed that the amplitude of the LPC effect in the facilitation condition (0.39 µV) was smaller than in spurious (2.43 µV, *p* = 0.039) and pure (2.32 µV, *p* = 0.045) interference conditions, which did not differ each other.

## 4. Discussion

In keeping with previous studies using the Stroop task, reaction times were slower for incongruent trials than for all the other conditions. Therefore, the meaning of the word affected the response to the colored stimuli in that it was impossible to ignore the word meaning although it was not relevant to the performance of the task [19]. Further, response times for neutral trials intermixed with incongruent trials were slower than those for neutral trials intermixed with congruent trials. This represents a novel finding, made possible by the unique task design of the present study. Indeed, leveraging the advantages of both block and intermixed designs, we were able to not only demonstrate the need of including a neutral condition as a control in all types of Stroop tasks but also highlight the crucial contribution of the task set to determine different performance between the more complex (incongruent plus neutral) and the simpler (congruent plus neutral) blocks, although neutral trials were provided similarly between the two. Further, previous work acknowledged the ’facilitation effect’ as the difference in reaction time between the congruent and neutral conditions [12,27]. Instead, the nomenclatures related to the RT difference between incongruent and neutral conditions, and between incongruent and congruent conditions, have generated misunderstandings, hampering proper conclusions among studies. The ‘Inhibition effect’ should refer to the former [27], while the ‘Interference effect’ should reflect the latter. However, the notion of the ‘Interference effect’ is also usually employed in the literature to designate the ‘Inhibition effect’ [40], thus generating huge inconsistencies among findings. Indeed, the RT difference between incongruent and congruent trials will always be “inflated” due to the sum of different processes subtending independent congruency effects: one related to the facilitation induced by congruency and one related to the interference generated by incongruence. For this reason, we have referred to this difference as “spurious interference”. Concerning the accuracy rates, we found that commission errors were higher for incongruent and lower for congruent than for both neutral conditions. On one side, these results confirm the increased difficulty in dealing with the interference generated by the conflict between the color word and ink color; on the other side, the improved accuracy in the congruent condition accounts for the facilitation generated by the correspondence of color word and ink color [41].

The issue of the relation between facilitation effects and Stroop interference has been consistently addressed [42,43,44]. Indeed, most theories at the root of the ‘color-word’ version of the Stroop task consider both effects because of the same mechanism, as a result of the congruency relations between the color name and the target color [44]. These models claim that the manipulation of congruency might influence the facilitation and the interference in parallel and should be explained by increased or decreased activity in those neural areas involved in response conflict detection, i.e., the anterior cingulate cortex [45]. However, other views challenged the existence of a common mechanism by showing the absence of a direct correlation between interference and facilitation effects [46], suggesting that they are driven by distinct, independent mechanisms. The analysis of the differential waveforms obtained from the relevant subtractions among conditions points to the identification of three main ERP activities labeled P3, N450, and LPC. Crucially, this paradigm allowed us to properly isolate the neural activity associated with each of the possible subtraction between conditions, thereby overcoming previous limitations due to misunderstanding the nomenclature.

The P3 effect was present in the pure interference and in the facilitation conditions but not in the spurious interference one. This finding shows that, at a relatively early stage of processing, a similar unfolding occurs for processes related to the word color compared to the neural words. These results relate to previous ERP studies [13,42,47,48] suggesting that, despite the tremendous difference in RT between incongruent and congruent conditions, at this timeframe (i.e., stimulus identification stage), the interference is not taking place. According to this view, the color and word are processed in parallel. Further, the absence of this component in the spurious interference condition reinforces the idea that conflict detection, reflected by the information processing in relation to the meaning of the stimulus, has not occurred yet. This is in line with longstanding literature claiming that, from a cognitive point of view, the P3 represents a complex and multi-factorial component, described as an index of the post-perceptual categorization process [49], as a measure of processing capacity [50] and associated with numerous processes related to response processing and task closure [51] but not with conflict monitoring processes.

The N450 effect was highest in the pure interference condition compared to the others. This result points to a stronger brain process for the incongruent condition, but only in the case of subtraction with the neutral condition. Indeed, the component is reduced only in the case of subtraction with the congruent condition. Previous literature consistently reports greater amplitudes of conditions of increased rather than reduced conflict [17,52,53]. Source analyses related to this component showed the critical role played by the anterior cingulate cortex (ACC) during conflict monitoring [15,40]. Liotti et al. [20] reported similar negativity in the analyses of congruent and incongruent condition subtraction waveforms at a latency of 410 ms over the midline area and interpreted it as evidence of conflict processing and resolution in the ACC. Similarly, West and Alain [17] showed a negative locus over bilateral frontocentral sites, which was more negative for incongruent than for congruent and neutral trials, interpreted as f conflict detection.

The occurrence of the LPC effect was limited to the interference conditions only. This finding suggests that, at a late stage of processing, the effects associated with congruency vanish but those related to the increased conflict generated by incongruency remain. Liotti et al. [20] reported a similar positivity in the 500–800 ms interval but limited to the left temporoparietal areas, which was interpreted as an extra word meaning analysis, i.e., after conflict resolution. According to West and Alain [17], the occurrence of a similar positive wave at similar latencies but peaking on left temporoparietal sites and stronger for the incongruent condition suggested extra ‘color pathway’ processing occurring to drive the proper response after the color-word incongruency had been recognized. The LPC effect might relate to the latter interpretation in that it occurred at a very late stage of processing, i.e., along response execution, and was evident only in the subtractions where the incongruence condition was the minuend. This component further relates to a recent study [54] in which the authors showed the occurrence of a late posterior positivity (LPP), which was enhanced in incongruent rather than congruent and neutral trials.

Overall, the present ERP findings suggest that the facilitation induced by the congruency and the interference induced by the incongruency represent independent mechanisms [48]. The observation of the waveform and the topography of the facilitation differential wave (i.e., Cong-Neu) do not point to the existence of a distinctive ERP limited to this specific subtraction. Indeed, the P3 emerged also after the subtraction of the neutral condition from the incongruent one (i.e., Incong-Neu), although the component was canceled out after subtracting the congruent condition from the incongruent one (i.e., Incong-Cong). Therefore, this component seems to reflect the increased cognitive effort required by task-relevant stimuli (i.e., word-color words) compared to neutral words. Further, the ERP interference effect, reflected by the N450 component, was higher when subtracting the neutral condition compared to the congruent condition from the incongruent one. However, for the very late stage of processing (i.e., LPC), the subtractions did show a similar unfolding. Crucially, the observed differences among conditions are in line with and extend previous findings highlighting specific modulations in the amplitude of distinctive preparatory slow waves underlying facilitation and interference effects [55].

One main limitation of the present study is related to the lack of a typical “switching” condition in which congruent, incongruent, and neutral trials were randomly intermixed. Future studies should adopt a switching design in order to better clarify how blocked presentations vs. switching presentations could influence the interference control processes. Further, since a recent ERP study using a visuomotor task acknowledged differences in performance between males and females [56], future investigations should address gender differences in the Stroop test. Another limitation of our study is related to the lack of correlations between behavioral performance and ERP markers of facilitation and interference. Future studies should include more participants and address in a more systematic way the relation between Stroop performance and ERP behavior.

## 5. Conclusions

In conclusion, in this study, we provided crucial electrophysiological evidence pointing at different brain mechanisms related to facilitation and interference effects occurring during a Stroop task. Further, we demonstrated the importance of building proper Stroop task designs to better dissociate the two effects and highlighted the need for future studies to include a proper neutral condition.

## Figures and Tables

**Figure 1 brainsci-13-00982-f001:**
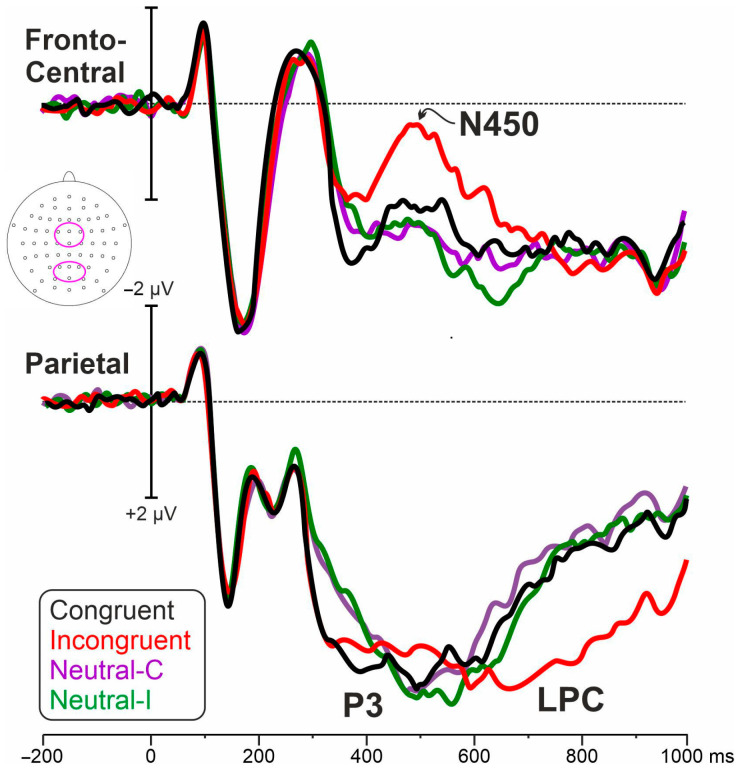
ERP waveforms in each of the four conditions. The top-view head schema shows the electrodes pooled in the ROIs.

**Figure 2 brainsci-13-00982-f002:**
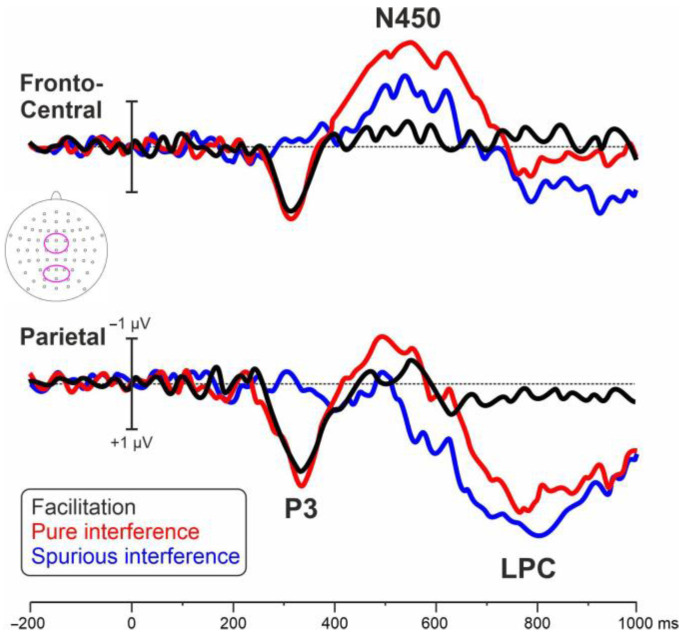
Differential waveforms. The Congruent minus neutral comparison represents the facilitation effect. The Incongruent minus neutral comparison represents the pure interference effect. The Incongruent minus congruent comparison represents the spurious interference effect. The top-view head schema shows the electrodes pooled in the ROIs.

**Figure 3 brainsci-13-00982-f003:**
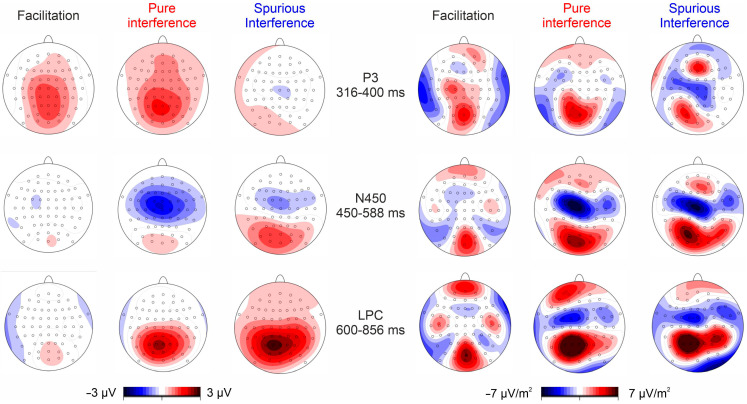
Scalp topography of differential ERP at the three studied intervals. The left panel shows the voltage distribution, while the right panel shows the current density distribution.

**Table 1 brainsci-13-00982-t001:** Behavioral results in terms of response time (RT) in milliseconds and commission errors percentage (CE%). Standard deviation (SD) is also reported.

	RT ± SD	CE% ± SD
Incongruent	576 ± 57	8.4 ± 6.4
Neutral-I	522 ± 36	6.0 ± 4.33
Congruent	504 ± 26	4.1 ± 3.6
Neutral-C	505 ± 35	6.1 ± 4.3

## Data Availability

The data presented in this study are available on request from the corresponding authors. The data are not publicly available due to internal regulations.

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
