# Peer review of "Time Course of Reactive Brain Activities during a Stroop Color-Word Task: Evidence of Specific Facilitation and Interference Effects"

_brainsci, 2023, doi:10.3390/brainsci13070982_

Round 1

Reviewer 1 Report

In this study, the authors aimed to provide new ERP evidence in order to test alternative hypotheses of the early (early parallel processing of word and color in the parallel distributed processing model) vs late (late selection response competition theories) locus of cognitive interference in the Stroop Colour and Word Test (SCWT).

The authors predicted that the usually reported Stroop components, i.e., the P3 (allocation of attentional resources during stimulus processing), the N450 (conflict detection) and the LPC (contextual integration and post-decision closure) should present a larger amplitude in the incongruent condition (conflicting condition) than in the congruent one (control condition).

Remark: consistently, it has been shown that a fronto-central N2 component is found to reflect conflict monitoring processes and overcoming of inhibition, with its main neural generator being the anterior cingulate cortex (ACC). Then, for cognitive control tasks that involve a linguistic component like the Stroop task, the N2 is followed by a centro-posterior N400 (inhibitory processes and interference suppression) and subsequently a late sustained potential (LSP ; discussed as reflecting either an engagement of executive processes (Hanslmayr et al., 2008), conflict resolution processes (Coderre et al., 2011; Heidlmayr et al., 2015; Naylor et al., 2012; West, 2004), semantic reactivation of the meaning of words following conflict resolution (Appelbaum et al., 2009; Liotti et al., 2000) or response selection (West, 2003, 2004)).

I would suggest the author to better specify the functional role of the ERP response generally used as marker of the different processes of executive functioning. I suggest the authors to rely on a recent review wrote by Heildmayr, Kihlsted, and Isel (2020, Brain and Cognition) in which the authors detailed these different ERP markers as a function of the executive task used.

2. Materials and Methods

2.1. Participants

Do the authors control the socioeconomic status (SES) of their participants? It was argued in the executive functioning literature that the SES could accounts for the performance of interference control in certain task involving executive functioning (see Bialystok, 2018, International Journal of Bilingual Education and Bilingualism).

Materials

It is not clear for me to what the “Congruent-Neutral” and “Incongruent-Neutral” runs refer? In the Stroop task, neutral condition is a condition I which non colour words are written with the colour used in both the congruent and incongruent situation. For example, DOG. Could the authors better explain this point.

Procedure

Were the Congruent and Incongruent trials presented in different blocks (blocked presentation)? If so, why this methodological choice was made? Do the authors think that the fact to have such blocked presentation could impact the interference control processes? How the switching between block could influence the interference control processes?

Results

What is “spurious interference” in Figure 2? It is not clear for me.

Weren't the authors expecting a larger amplitude of the P300 in incongruent (red) than in congruent (black) trials? If it is the case, descriptively, Figure failed to show such an attentional effect for the incongruent condition. How the authors explain this lack of P300 effect?

The English language is good.

Author Response

In this study, the authors aimed to provide new ERP evidence in order to test alternative hypotheses of the early (early parallel processing of word and color in the parallel distributed processing model) vs late (late selection response competition theories) locus of cognitive interference in the Stroop Colour and Word Test (SCWT).
The authors predicted that the usually reported Stroop components, i.e., the P3 (allocation of attentional resources during stimulus processing), the N450 (conflict detection) and the LPC (contextual integration and post-decision closure) should present a larger amplitude in the incongruent condition (conflicting condition) than in the congruent one (control condition).
Remark: consistently, it has been shown that a fronto-central N2 component is found to reflect conflict monitoring processes and overcoming of inhibition, with its main neural generator being the anterior cingulate cortex (ACC). Then, for cognitive control tasks that involve a linguistic component like the Stroop task, the N2 is followed by a centro-posterior N400 (inhibitory processes and interference suppression) and subsequently a late sustained potential (LSP ; discussed as reflecting either an engagement of executive processes (Hanslmayr et al., 2008), conflict resolution processes (Coderre et al., 2011; Heidlmayr et al., 2015; Naylor et al., 2012; West, 2004), semantic reactivation of the meaning of words following conflict resolution (Appelbaum et al., 2009; Liotti et al., 2000) or response selection (West, 2003, 2004)).
I would suggest the author to better specify the functional role of the ERP response generally used as marker of the different processes of executive functioning. I suggest the authors to rely on a recent review wrote by Heildmayr, Kihlsted, and Isel (2020, Brain and Cognition) in which the authors detailed these different ERP markers as a function of the executive task used.

Response: We thank the reviewer for this interesting comment. Indeed, there is a large variety of ERPs studies investigating the Stroop effect and, as largely acknowledged in the EEG community, the occurrence of ERP components is mostly related to the specific task used and to the overall experimental designs. In our study, we used an innovative approach in that we differentiated the facilitation and the interference effects (usually “confounded” in the common version of the task intermixing congruent and incongruent trials) leveraging a block design separating congruent and incongruent trials. Since no previous investigation used the same design, we hypothesized the occurrence of some ERP components acknowledged by similar ERP studies (e.g.) Liotti, Woldorff, Perez, Mayberg (2000), but we did not exclude an exploratory approach due to the novelty of our manipulations. However, we thank the reviewer for the interesting suggestions related to the centro-posterior N400, the late sustained potential (LSP), and the recent review by Heildmayr and co-authors (2020). We have added the cited references in the revised version of the manuscript better specifying the functional role of the ERP components.

  1. Materials and Methods
2.1. Participants
Do the authors control the socioeconomic status (SES) of their participants? It was argued in the executive functioning literature that the SES could accounts for the performance of interference control in certain task involving executive functioning (see Bialystok, 2018, International Journal of Bilingual Education and Bilingualism).
Response: As suggested, we now checked the participants’ socioeconomic status and found that was similar. We have clarified this issue in the revised version of the manuscript.
Materials
It is not clear for me to what the “Congruent-Neutral” and “Incongruent-Neutral” runs refer? In the Stroop task, neutral condition is a condition I which non colour words are written with the colour used in both the congruent and incongruent situation. For example, DOG. Could the authors better explain this point.
Response: We apologize with the reviewer for not being clear on this point. With the nomenclature “Congruent-Neutral” and “Incongruent-Neutral” runs we refer to the fact that, in each block, the neutral trials were intermixed with congruent and incongruent trials. We have clarified this issue in the revised version of the manuscript.
Procedure
Were the Congruent and Incongruent trials presented in different blocks (blocked presentation)? If so, why this methodological choice was made? Do the authors think that the fact to have such blocked presentation could impact the interference control processes? How the switching between block could influence the interference control processes?

Response: We thank the reviewer for the comment, which allows us to better clarify the choice of this design. In a typical Stroop task, intermixing congruent and incongruent trials, it has been repeatedly shown that the recognition of the word is highly facilitated in the case of congruent trials but is highly hampered in the case of incongruent trials. Further, it is common to use a neutral condition as a “reference” in which no facilitation or interference occurs, thus allowing a better comparison between conditions. The advantage of our design relies upon the fact that we could “elegantly” dissociate the subprocess related to the facilitation from those related to the interference and to have obtained a “pure” control condition (the respective neutral trials) for each process. Relatedly, note that the reaction times for the neutral trials (which were identical in the two blocks) were different between the incongruent-neutral and the congruent-neutral blocks being slower in the former. This confirms that neutral trials were indeed influenced at a block level, demonstrating that facilitation and interference were occurring with a certain strength in the two blocks. Therefore, the evidence that the two sub-processes were “visible” at the behavioral level made us sure that these could be reflected at the neural level, and then captured by ERPs. Of course, we believe that such blocked presentation had a different impact in relation to interference control processes and we have also provided evidence in favor of this hypothesis (see comparisons of ERPs and scalp topographies of pure vs. spurious interference for instance). However, we do not exclude that a similar trend would have emerged in a typical “switching” condition. Relatedly, a recent study by Perri and co-authors used a switching Stroop test in hypnotized subjects (Perri, Bianco, Facco & Di Russo (2021) and identified an N300 and an LPP. In the revised version of the manuscript, we have added a small section highlighting the absence of a switching condition as a limitation of the study and calling for further investigations.

Results
What is “spurious interference” in Figure 2? It is not clear for me.

Response: As stated in the discussion section, there are inconsistencies among findings related to the ‘real’ interference effect in the Stroop test. Indeed, some studies refer to the interference as the difference between incongruent and congruent conditions while other studies refer to the interference as the difference between incongruent and neutral conditions. However, the former (i.e., the difference between incongruent and congruent conditions) will always be “inflated” due to the sum of different processes subtending independent congruency effects: one related to the facilitation induced by congruency and one related to the interference generated by incongruence. For this reason, we have referred to this difference as ‘spurious interference’.

Weren't the authors expecting a larger amplitude of the P300 in incongruent (red) than in congruent (black) trials? If it is the case, descriptively, Figure failed to show such an attentional effect for the incongruent condition. How the authors explain this lack of P300 effect?

Response: The reviewer is correct in highlighting that the P3 was similar between congruent and incongruent trials. Indeed, this component seems to reflect the increased cognitive effort required by task-relevant stimuli (i.e., word-color words) compared to neutral words and is not related to conflict monitoring. This is in line with longstanding literature claiming that, from a cognitive point of view, the P3 represents a complex and multi-factorial component, described as an index of the post-perceptual categorization process (e.g. Mecklinger and Ullsperger,1993), as a measure of processing capacity (e.g., Kok, 2001) and associated to numerous processes related to response processing and task closure (e.g., Soltani & Knight 2000). We have acknowledged this in the discussion section of the revised manuscript.

Reviewer 2 Report

Overall authors have done good attempt .

How ever authors did not state if the gender difference will make any impact to the study as the authors have taken adults (both male and female)

Authors must also include the selection criteria very specifically to nullify the impact of various confounding factors. 

Language quality is overall good. 

Author Response

Overall authors have done good attempt .
However authors did not state if the gender difference will make any impact to the study as the authors have taken adults (both male and female)

Response: We thank the reviewer for the comment. Indeed, in a previous study using a go/no-go task, we showed that males favored speed over accuracy while females favored accuracy over speed. Further, differences in specific ERPs amplitude were associated with behavioral performance.  In the revised version of the manuscript, we have added a paragraph calling for further investigations addressing gender differences in the Stroop task.

Authors must also include the selection criteria very specifically to nullify the impact of various confounding factors

Response: We apologize for not having specified the inclusion and exclusion criteria. We have added information concerning age requirements, right-handedness, socioeconomic status, and refraining from psychotropic medications in the revised version of the manuscript.

Reviewer 3 Report

The goal of the study is interesting, and the introduction highlights a number of intriguing limitations on the topic, but the expectation set wasn't met because the design, methods, and results presented don't deepen or implement the conventional analyses already presented in other studies. Instead, they could have investigated additionally the correlations and predictions of the behavioral data with the behavior of the EEG signal, better validating the subtleties of the steps of the Stroop color-word task I believe the suggested study lacked a more pertinent and distinctive methodology.

It is ok

Author Response

The goal of the study is interesting, and the introduction highlights a number of intriguing limitations on the topic, but the expectation set wasn't met because the design, methods, and results presented don't deepen or implement the conventional analyses already presented in other studies. Instead, they could have investigated additionally the correlations and predictions of the behavioral data with the behavior of the EEG signal, better validating the subtleties of the steps of the Stroop color-word task I believe the suggested study lacked a more pertinent and distinctive methodology.

Response: We thank the reviewer for the comment, and we understand the concerns. We agree that addressing the correlations and predictions of the behavioral data with ERPs might have deepened the conventional analysis used in other studies. However, this was not the aim of our study, which was instead to explore ERP markers of facilitation and interference mechanisms of the Stroop test. We performed exploratory correlation analyses between behavioral performance and ERP amplitude, but we did not find any significant trend. The reason of the lack of correlations might stem from the limited number of participants, i.e., 18. However, we performed a power analysis on the basis of the main aim of our study. Of course, including more participants could be more suitable to unveil possible correlations between performance and ERP behavior and we agree that this is a limitation of our study. Therefore, in the revised version of the manuscript we have added a limitation section in which we have called for future studies to include more participants and address the association between behavioral performance and ERP markers.

Round 2

Reviewer 1 Report

I thank the authors for the quality of the revised version of the article in a short period. I appreciate that my commentaries was considered to improve the manuscript.

Author Response

We thank again the Reviewer for the work done on the manuscript.

Reviewer 3 Report

The authors responded to the comments appointed and justified the no performed the additional analysis in the limitation item included in the conclusion, but I suggest moving this paragraph to the end of the discussion section.  I assess the new article version as adequate to publish.  

It is ok

Author Response

Thank you for the suggestion. We now moved the limitation paragraph to the end of the discussion section.